# Autologous P63+ lung progenitor cell transplantation in idiopathic pulmonary fibrosis: a phase 1 clinical trial

Shiyu Zhang[1,2†], Min Zhou[3†], Chi Shao[4†], Yu Zhao[1,2†], Mingzhe Liu[1,2], Lei Ni[3], Zhiyao Bao[3], Qiurui Zhang[3], Ting Zhang[5], Qun Luo[6*], Jieming Qu[3*], Zuojun Xu[4*], Wei Zuo[1,2,5,7*]

[1]Shanghai East Hospital, School of Medicine, Tongji University, Shanghai, China; [2]Tongji Stem Cell Center, Tongji University, Shanghai, China; [3]Department of Respiratory and Critical Care Medicine, Ruijin Hospital, Shanghai Jiao Tong University School of Medicine, Shanghai, China; [4]Department of Respiratory and Critical Care Medicine, Peking Union Medical College Hospital, Chinese Academy of Medical Sciences & Peking Union Medical College, Beijing, China; [5]Super Organ R&D Center, Regend Therapeutics, Shanghai, China; [6]Department of Respiratory and Critical Care Medicine, The First Affiliated Hospital of Guangzhou Medical University, Guangzhou, China; [7]Kiangnan Stem Cell Institute, Zhejiang, China

*For correspondence:
luoqunx@163.com (QL);
jmqu0906@163.com (JQ);
xuzj@hotmail.com (ZX);
zuow@tongji.edu.cn (WZ)

†These authors contributed equally to this work

## eLife Assessment

This **important** study describes a first-in-human trial of autologous p63+ stem cells in patients with idiopathic pulmonary fibrosis, a lethal condition for which effective treatments are lacking. The authors provide **convincing** evidence that P63+ progenitor cell therapy can be safely delivered in patients with ILD, warranting movement to a Phase 2. However, given that this is a Phase 1 study with a small sample size, conclusions regarding efficacy should not yet be made.

## Abstract

**Background:** In idiopathic pulmonary fibrosis (IPF) patients, alveolar architectures are lost and gas transfer function would decline, which cannot be rescued by conventional anti-fibrotic therapy. P63+ lung basal progenitor cells are reported to have potential to repair damaged lung epithelium in animal models, which need further investigation in clinical trials.

**Methods:** We cloned and expanded P63+ progenitor cells from IPF patients to manufacture cell product REGEND001, which were further characterized by morphology and single-cell transcriptomic analysis. Subsequently, an open-label, dose-escalation autologous progenitor cell transplantation clinical trial was conducted. We treated 12 patients with ascending doses of cells: 0.6x, 1x, 2x and 3.3x$10^6$ cells/kg bodyweight. The primary outcome was the incidence and severity of cell therapy-related adverse events (AEs); secondary outcome included other safety and efficacy evaluations.

**Results:** P63+ basal progenitor cell was safe and tolerated at all doses, with no dose-limiting toxicity or cell therapy-related severe adverse events observed. Patients in three higher dose groups showed significant improvement of lung gas transfer function as well as exercise ability. Resolution of honeycomb lesion was observed in patients of higher dose groups.

**Conclusions:** REGEND001 has high safety profile and meanwhile encourages further efficacy exploration in IPF patients.

**Funding:** National High Level Hospital Clinical Research Funding (2022-PUMCH-B-108), National Key Research and Development Plan (2024YFA1108900, 2024YFA1108500), Jiangsu Province Science and Technology Special Project Funding (BE2023727), National Biopharmaceutical Technology Research Project Funding (NCTIB2023XB01011), Non-profit Central Research Institute Fund of Chinese Academy of Medical Science (2020-PT320-005), and Regend Therapeutics.
**Clinical trial number:** Chinese clinical trial registry: CTR20210349.

## Introduction

Idiopathic pulmonary fibrosis (IPF) is a lethal disease of unknown etiology, characterized by irreversible alveolar damage and lung tissue fibrosis. The progression of IPF would lead to persistent decline of lung function, attenuated exercise ability, impaired quality of life and eventually death within 3~5 years after disease onset (*Raghu et al., 2022*). Currently, there is no cure for IPF, and the clinically applied treatments, such as self-care measures, anti-fibrotic medication, and pulmonary rehabilitation, can only help relieve the symptoms and/or slow down the progression of IPF. Among them, the first-line anti-fibrotic drugs Nintedanib and Pirfenidone could only reduce the rate of decline in lung function (primarily lung forced vital capacity) rather than halt or reverse the disease progression. Furthermore, patients with IPF may discontinue Nintedanib and Pirfenidone due to adverse events such as nausea and diarrhea (*Vancheri et al., 2018*; *Flaherty et al., 2018*; *Milger et al., 2015*; *Richeldi et al., 2014*; *Noble et al., 2011*). More importantly, none of these conventional medications could repair or regenerate the injured lung tissues in IPF patients. Therefore, multiple pluripotent stem cell or adult stem/progenitor cell-based regenerative strategies are being widely studied at the pre-clinical or clinical level for IPF treatment (*Cheng and Ghodsi, 2024*; *Herriges et al., 2023*; *Shi et al., 2019*; *Antoniou et al., 2017*).

Following lung injury, a number of tissue-specific stem cells, including airway basal progenitor cells, secretory progenitor cells and type 2 alveolar stem cells, are responsible for lung epithelial repair and regeneration (*Basil et al., 2024*). Among these cell populations, airway basal stem/progenitor cells marked by the expression of TP63 (P63), keratin 5 (KRT5) have been widely studied (*Zuo et al., 2015*; *Vaughan et al., 2015*; *Kumar et al., 2011*). These P63+ progenitor cells located in the airway basal layer are known to be capable of differentiating into secretory, goblet and ciliated cells and reconstituting the airway epithelium (*Davis and Wypych, 2021*). In the recent decade, it has been discovered that these P63+ progenitor cells in airway can be activated by multiple types of major alveolar injuries (eg. IPF, influenza infection, COVID-19), and then migrate from airway to the damaged loci in the alveolar compartment, where they play complex roles in the pathogenesis process (*Zuo et al., 2015*; *Vaughan et al., 2015*; *Heydemann et al., 2023*; *Prasse et al., 2019*). It is proposed that the normal P63+ progenitor cells in the alveolar area are able to expand and rapidly form epithelial barriers to 'band-aid' the injured lung tissue (*Kumar et al., 2011*; *Zacharias et al., 2018*). When stimulated by proper microenvironment signals, the P63+ progenitor cells could also differentiate towards alveolar epithelium lineages to facilitate lung repair and regeneration (*Zuo et al., 2015*; *Vaughan et al., 2015*; *Xi et al., 2017*; *Kathiriya et al., 2020*). However, in the context of IPF, it has been confirmed that there are some dysplastic P63+ progenitor cells distributed near the fibrotic foci, which might be pro-fibrotic with potential to exacerbate the IPF disease (*Habermann et al., 2020*; *Jaeger et al., 2022*; *Wang et al., 2023*). Therefore, in IPF patients, the function of normal or dysplastic P63+ progenitor cells still need to be investigated in details especially in clinical trials.

Previous studies had evaluated the therapeutic potential of mouse/human P63+ progenitor cell transplantation in a bleomycin-injured pulmonary fibrosis murine model. The transplanted mouse/human P63+ progenitor cells were able to engraft into the injured mouse lung tissue, re-establish the epithelial barrier, reconstitute vascularized air sac, and improve the blood oxygen levels of the mice. Transplantation of P63+ progenitor cells could also attenuate collagen deposition and myofibroblast cell proliferation in lung and eventually reduce mouse mortality (*Shi et al., 2019*; *Zhou et al., 2020*; *Zhou et al., 2021*). Pre-clinical safety evaluations were also performed on both mice and monkeys following Good Laboratory Practice guidelines, showing that intrapulmonary transplantation of P63+ progenitor cells is very safe (*Zhou et al., 2021*). In a pilot clinical study conducted in 2016, two patients with severe non-cystic fibrosis bronchiectasis were treated with autologous P63+ progenitor cell transplantation. Pulmonary functions of both patients recovered remarkably, with none aberrant cell

growth or other related adverse events (AEs) during the whole follow-up time (*Ma et al., 2018*). In a very recent report of a controlled phase 1 clinical trial, autologous P63+ progenitor cell transplantation was performed to treat 17 patients with chronic pulmonary obstructive disease (COPD). These data indicated that the cell treatment could significantly improve the gas transfer function (DLCO) of the COPD lung (*Wang et al., 2024*). Altogether these previous pre-clinical and clinical evidences encourage us to further test the cell therapy in the pulmonary fibrosis disease at clinical stage.

Here, we report the first clinical trial treating IPF patients with autologous P63+ lung progenitor cell product (REGEND001) transplantation. The progenitor cells were isolated from a trace amount of healthy airway epithelium sample obtained via bronchoscopic brushing. After 3–5 weeks of cell expansion, up to 100 million P63+ lung progenitor cells were produced and transplanted into the lungs of IPF patients via bronchoscopy. The safety and potential efficacy of this novel cell therapy for treating IPF were evaluated in this clinical trial.

## Methods

### Study design

An open-label, dose-escalation, phase I clinical trial (CTR20210349, translated English version in Supplementary Material) was conducted at three clinical research centers (Peking Union Medical College Hospital, Shanghai Ruijin Hospital, The First Affiliated Hospital of Guangzhou Medical University) in China to evaluate the safety, tolerability, and efficacy of treating IPF patients through autologous P63+ basal progenitor cell transplantation at different dosages. This trial was approved by the National Medical Products Administration and the ethic committee of the three clinical research centers, and was conducted in accordance with the Declaration of Helsinki and the principles of Good Clinical Practice (GCP).

### Participants

Eligible patients were adults between the ages of 50 and 75 who were diagnosed with IPF based on the 2018 guidelines for diagnosis of IPF by the Pulmonary Pathology Society. Additional eligibility criteria were a DLCO that was 30 to 79% of the predicted value, a FVC that was 50% or more of the predicted value in pulmonary function tests within the preceding 3 months of patients screening. Classic radiological features of IPF were observed from their high-resolution computed tomography (HRCT) in the previous 12 months and they were tolerable with bronchofiberscopy. Full inclusion and exclusion criteria are listed in Appendix 1. Prior to participating in the study, all participants were provided with detailed information regarding the study objectives and design, and informed consent was obtained. Patients were assigned to receive a single administration of autologous P63+ lung progenitor cells at $0.6 \times 10^6$, $1 \times 10^6$, $2 \times 10^6$, or $3.3 \times 10^6$ cells/kg bodyweight. All patients received standard IPF treatment during the study period. For more details, please see full clinical trial protocol in Supplementary Materials.

### Clone P63+ airway basal progenitor cells

To isolate human airway basal progenitor cells, the brushing samples were collected from the 3rd to 5th level bronchi tissue of donors. P63+ progenitor cells are selectively expanded through a pharmaceutical-grade cell cloning culture system. 50 µg/mL gentamicin sulfate is applied at the initial passage but not in the subsequent 2–5 passages to prevent microbial contamination. Each batch of cell products were confirmed to be free of bacterial and mycoplasma contamination, or endotoxin, BSA, and antibiotic residues. After 3–4 passages, airway basal progenitor cell clones were subjected to immunofluorescence staining for KRT5 and P63. The full details of methods to expand P63+ lung progenitor cells have been described by Frank McKeon/Wa Xian group (*Rao et al., 2020*), which is adapted to pharmaceutical-grade technology patented by Regend Therapeutics, Ltd.

### Single-cell RNA sequencing and bioinformatics

After removing feeder cells through gradient digestion, the passage 0 primary cells isolated by the pharmaceutical-grade cell cloning culture system were digested to prepare a single-cell suspension. Single cells were captured and barcoded in 10×Chromium Controller (10×Genomics). Subsequently, RNA from the barcoded cells was reverse-transcribed and sequencing libraries were prepared using

Chromium Single-Cell 3'v3 Reagent Kit (10×Genomics) according to the instructions of manufacturer. Sequencing libraries were loaded on an Illumina NovaSeq with 2×150 paired-end kits at Novogene, China. Raw sequencing reads were processed using the Cell Ranger v.3.1.0 pipeline from 10×Genomics. In brief, reads were demultiplexed, aligned to the human GRCh38 genome, and UMI counts were quantified per gene per cell to generate a gene-barcode matrix. Post-processing, including filtering by number of genes and mitochondrial gene content expressed per cell, was performed using the Seurat v.4.3.0 (*Hao et al., 2024*). Genes were filtered out that were detected in less than three cells. A global-scaling normalization method 'LogNormalize' was used to normalize the data by a scale factor (10000). Next, a subset of highly variable genes was calculated for downstream analysis and a linear transformation (ScaleData) was applied as a pre-processing step. Principal component analysis (PCA) dimensionality reduction was performed with the highly variable genes as input in Seurat function RunPCA. The top 30 significant PCs were selected for two-dimensional Uniform Manifold Approximation and Projection (UMAP), implemented by the Seurat software with the default parameters. FindCluster in Seurat was used to identify cell clusters. Marker genes were identified through differential expression analysis utilizing the FindAllMarkers function in Seurat. Genes showing differential expression, observed in at least 25% of cells within the cluster, and exhibiting a fold change greater than 0.25 on a logarithmic scale, were classified as marker genes. Gene Ontology (GO) enrichment analysis of differentially expressed genes was implemented by the ClusterProfiler R package. GO terms with p-value <0.05 were considered significantly enriched. Dot plots were used to visualize enriched terms by the enrichplot R package (*Yu et al., 2012*).

## Cell morphology analysis

P63+ progenitors were selectively expanded and cultured to the P4 generation, and photographs of the cell clones were taken at 10 x magnification. ImageJ (15.2 a) was used to perform cell morphology analysis of progenitor cells from each patient, with roundness parameter employed to describe the morphology of cell clones. Pearson's correlation scores were calculated to assess the relationship between the cell morphology and the age as well as lung function index. Statistical significance was considered at a p-value <0.05.

## Interventions

To sample autologous airway tissue, the bronchoscopy was performed by a team of certified physicians using a flexible fiberoptic bronchoscope. Before the bronchoscopy, 2% lidocaine spray anesthesia or general anesthesia was used. Peking Union Medical College Hospital and Shanghai Ruijin Hospital used spray anesthesia, while The First Affiliated Hospital of Guangzhou Medical University used general anesthesia. The total dosage of lidocaine was limited to 8.2 mg/kg bodyweight. During bronchoscopy, the patient is usually placed in a supine position. To qualify for tissue sampling, the surface of the airway should be smooth with normal color under bronchoscopy, with no new growths, ulcers, or bleeding points. Afterwards, a small amount of tissue sample was collected from the 3-5th level bronchi of IPF patients by sliding the brushes tenderly. Brushed samples were then shipped from hospital to cell factory through cold-chain transport (2–8°C) within 48 hr. In the cell factory, single-cell suspension was prepared by washing and enzymatically digestion, which were then cultured under a patented regenerative cell cloning (R-Clone) system (*Wang et al., 2024*). The final product passed a series of rigorous tests, including identity, purity, sterility, residual endotoxin, viral contamination, residual BSA and residual antibiotics, etc., were qualified for this clinical trial. The qualified autologous P63+ lung progenitor cells product would be suspended in 14 mL saline, sealed in a cell preservation bag and shipped freshly to the hospital through cold-chain transport (2–8°C) within 48 hr.

To prepare for autologous P63+ lung progenitor cells transplantation, the cell suspension was re-suspended in 28 mL saline and then averagely injected into bronchial segments in middle and lower lobes (3 mL each) through the fiberoptic bronchoscope. All patients maintained a supine position for 2 hr, avoided drinking and minimized coughing for 3 hr post administration. When necessary, oral codeine was given.

In principle, all patients continued to use medications they had taken before participating this clinical trial, except Nintedanib. Nintedanib was excluded from the allowed medication list because as tyrosine kinase inhibitor, it is worried to have growth inhibition effect on the transplanted progenitor cells. Compatible concomitant IPF medications include Pirfenidone, Omeprazole, N-Acetyl Cysteine,

Methylprednisolone, and Prednisone. The details of all concomitant medications were recorded on the Concomitant Drug Use page of the Case Report Form, clarifying the reason for medication/treatment, administration/treatment methods, and start and end dates.

## Outcomes

The primary outcome of the study was the incidence and severity of the cell therapy-related AEs within 24 weeks after treatment. AEs were defined as abnormal laboratory results, symptoms, or signs, and were graded using the Common Terminology Criteria for Adverse Events (Version 5.0). The association between AEs and cell administration was evaluated by pulmonologists. Physician assessments, clinical laboratory evaluations (including complete blood count, serum biochemistry tests, and ECGs), and follow-up assessments were recorded for all patients with AEs, with treatment given when necessary.

The secondary outcome measures included incidence of complication related to bronchoscopy, examination of blood routine, urine routine, blood biochemistry, 12-lead Electrocardiogram (ECG) and IPF exacerbation events within 24 weeks after treatment; examination of Carcinoembryonic antigen (CEA), Neuron-specific enolase (NSE), Cytokeratin-19-fragment (CYFRA21-1) and Squamous cell carcinoma antigen (SCC) at baseline, 12 weeks and 24 weeks after treatment; evaluation of cell therapy efficacy through DLCO-SB, FVC, DLCO/VA, exercise tolerance test (6MWD), quality-of-life questionnaire SGRQ at 4 weeks, 12 weeks, and 24 weeks after treatment; change of imaging of lung by HR-CT at 24 weeks after treatment. The quality of lung function test would be evaluated according to European Respiratory Society (ERS)/ American Thoracic Society (ATS) guidlines (*Graham et al., 2017*). For more details, please see full clinical trial protocol in the Supplementary Materials and Methods.

## Statistical methods

All analyses were performed using R version 4.2.1 and GraphPad Prism version 9.0. p-value <0.05 was considered statistically significant. The n numbers for each experiment are provided in the text or figures. Descriptive statistics were used to summarize adverse events. Continuous data were presented as mean or median, and categorical data was presented as absolute numbers and percentages of patients in each category.

The statistical significance (p-value) of area under the curve (AUC) between the low and high dose groups for % predicted DLCO/VA, % predicted FVC and 6MWD were determined by an unpaired t test after normality validation with Shapiro-Wilk test, and the significance between baseline and 24-week data in the high dose group for DLCO, % predicted FVC and 6MWD, were determined by an paired t test after normality validation with Shapiro-Wilk test, as shown in Figure 3. The AUC between the low and high dose groups for DLCO and the significance between baseline and 24-week data in high dose group for % predicted DLCO/VA were analyzed using the Mann-Whitney U test following normality validation with Shapiro-Wilk test. p Values for continuous data in *Table 1* were calculated using one-way analysis of variance (ANOVA) followed by Tukey's post hoc for comparisons among four dose groups, except for Viability Rate, which was evaluated using the Kruskal-Walli's test followed by Dunn's post hoc test, after normality validation using the Shapiro-Wilk test. Categorical variables, such as sex, pre-study medication and concomitant medication, were compared by the Chi-square test or Fisher's exact test. DLCO data were presented as both absolute value (mmol/min/kPa) and percentage of predicted value (%). DLCO/VA data were presented as both absolute value (mmol/min/kPa/L) and percentage of predicted value (%). FVC data were presented as both absolute value (L) and percentage of predicted value (%). DLCO, DLCO/VA, FVC, SGRQ, and 6MWD were presented as means.

## Results

### Enrollment of patients

We conducted an open-label, dose-escalation phase 1 clinical trial (CTR20210349) from July 19, 2021 to June 9, 2023, to study the safety and efficacy of autologous P63+ lung progenitor cell transplantation in patients with IPF. Appendix 1 showed detailed patient inclusion and exclusion information.

**Table 1.** Baseline demographic and clinical characteristics.

| Dose | 0.6 M | | | 1 M | | | 2 M | | | 3.3 M | | | p-value |
|---|---|---|---|---|---|---|---|---|---|---|---|---|---|
| Patient ID | #103 | #105 | #106 | #201 | #303 | #305 | #108 | #203 | #306 | #111 | #112 | #204 | |
| Demographics | | | | | | | | | | | | | |
| Age (years) | 75 | 62 | 68 | 57 | 63 | 60 | 59 | 67 | 73 | 75 | 59 | 72 | 0.6916 |
| Sex | Male | Male | Male | Male | Male | Female | Male | Male | Male | Male | Male | Male | 1.0000 |
| Body mass index (kg/m²) | 22.91 | 26.04 | 23.07 | 23.93 | 22.27 | 21.49 | 20.69 | 28.39 | 24.42 | 26.93 | 26.44 | 24.15 | 0.2334 |
| Cell characteristics | | | | | | | | | | | | | |
| Viability Rate (%) | 94.00 | 91.00 | 95.50 | 97.00 | 97.00 | 98.00 | 99.00 | 99.00 | 97.00 | 98.00 | 96.00 | 97.00 | 0.0503 |
| Biological Efficacy (%) | 59.00 | 36.00 | 44.00 | 57.00 | 47.00 | 78.00 | 48.00 | 86.00 | 76.00 | 56.00 | 40.00 | 64.00 | 0.4858 |
| IPF characteristics | | | | | | | | | | | | | |
| DLCO (%predicted) | 50.70 | 49.32 | 56.64 | 46.48 | 52.17 | 44.90 | 31.41 | 40.60 | 33.29 | 44.03 | 70.38 | 60.72 | 0.8516 |
| DLCO/VA (%predicted) | 97.00 | 109.60 | 72.60 | 57.10 | 81.30 | 61.90 | 54.00 | 49.80 | 47.60 | 59.50 | 102.00 | 95.00 | 0.4315 |
| FVC (%predicted) | 68.60 | 54.10 | 104.40 | 98.37 | 71.20 | 74.60 | 72.00 | 112.91 | 59.50 | 70.00 | 90.00 | 74.69 | 0.8790 |
| Pre-study medication for IPF | | | | | | | | | | | | | |
| Pirfenidone | √ | - | √ | √ | √ | - | √ | √ | √ | √ | √ | - | 1.0000 |
| Nintedanib | - | - | - | - | - | - | - | - | √ | - | √ | - | 1.0000 |
| Acetylcysteine | - | - | - | √ | √ | - | √ | - | - | - | √ | √ | 0.5909 |
| Tiotropium Bromide | - | - | - | - | - | - | - | - | - | - | - | √ | 1.0000 |
| Concomitant medication for IPF | | | | | | | | | | | | | |
| Pirfenidone | √ | - | √ | √ | √ | - | √ | √ | √ | √ | √ | - | 1.0000 |
| Acetylcysteine | - | - | - | √ | √ | - | √ | - | - | - | √ | √ | 0.5909 |

Totally 12 patients were enrolled in the trial and assigned to one of four dose groups sequentially: 0.6 million cells/kg bodyweight (0.6 M), 1 million cells/kg bodyweight (1 M), 2 million cells/kg bodyweight (2 M), and 3.3 million cells/kg bodyweight (3.3 M; *Figure 1—figure supplement 1*). After the 3rd patient of the 3.3 M group was recruited, enrollment of patients was terminated. Baseline demographic data (*Table 1*) showed that the trial participants in all groups were generally matched. Of note, 75% participants used anti-fibrotic drug pirfenidone as concomitant medication in this trial. None of them used Nintedanib as concomitant medication.

## Clone P63+ progenitor cells from normal lung area of IPF patients

First, a small tissue sample is collected via bronchoscopy from grade 3–5 bronchi of upper lung lobes in patients with IPF. P63+ progenitor cells are selectively expanded through a pharmaceutical-grade cell cloning culture system. 50 µg/mL gentamicin sulfate is applied at the initial passage but not in the subsequent 2–5 passages to prevent microbial contamination. Each batch of cell products were confirmed to be free of bacterial contamination, or endotoxin, BSA, and antibiotic residues (*Figure 1A*). In IPF patients, the fibrotic foci are mainly distributed in lower or middle lobes. We selected relatively healthy lung upper lobes to collect tissue samples, which allowed us to collect the normal autologous P63+ progenitor cells instead of those pro-fibrotic cells (*Figure 1B*). Expression of representative progenitor cell markers KRT5 and P63 was confirmed by immunofluorescence staining and flow cytometric analysis demonstrating KRT5+ cell purity >95% (*Figure 1C*, *Figure 1—figure supplement 2A*). For each batch of cell product, no tumorigenic potential was confirmed in the soft agar colony formation assay (*Figure 1—figure supplement 2B*).

We noticed that the cell cloned from different individual patients exhibited diverse morphology: some clones were regularly round while others exhibited irregular shape (*Figure 1D*). To further analyze the individual variances in P63+ progenitors isolated from the 12 IPF patients, we conducted cell morphological analysis on P4 passages of all subjects and correlated them with patient age and lung function. The data revealed significant negative correlation between the age of patients and P63+ progenitor cell clone roundness, which might be related to previous findings that aging can negatively impact cell properties (*Moreno-Valladares et al., 2023*; *Figure 2A*). Interestingly, we also observed a positive correlation between P63+ progenitor cell clone roundness and the pulmonary function indexes DLCO, FVC, and FEV1 (*Figure 2A*).

To gain molecular insights into the identity composition of the cells selectively grown in the culture system, we performed single-cell sequencing on the cells. Unsupervised clustering of the cells uncovered three distinct subpopulations:C1, C2, and C3 (*Figure 2B*). The C1 population was characterized by lung progenitor classic markers KRT5 and P63 and accounts for more than half of all cells (65.03%; *Figure 2B and C*). Gene ontology analysis of differentially expressed genes showed that C1 cells demonstrated high expression of genes involved in Wnt Signaling pathway and negative regulation of apoptotic signaling pathway, which is crucial for maintaining the identity of stem cells and the development of the lung (*Frank et al., 2016*). Notably, C1 cells also demonstrate the function of positive regulation of angiogenesis and negative regulation of fibroblast proliferation (*Figure 2D*). The C2 populations accounts for 23.40% of the total number of cells. It also expressed KRT5, P63, and meanwhile expresses the proliferation-related marker KI67 (*Figure 2B and C*). C2 was mainly enriched in mitosis-related genes, which is consistent with its identity as a proliferative state (*Figure 2D*).

C3 cells accounts for 11.57% of the total number of cells, which showed very low level of KRT5 or P63 expression, but expressed high level of alveolar epithelial cell types 1 (AEC1) gene HOPX. Of note, the C3 cells also expressed a set of genes known as markers for variant progenitor cells, including CXCL17, CEACAM6, IL1RN, and CLDN4 (*Figure 2E*; *Wang et al., 2023*). Further gene ontology analysis identified differentially expressed genes in C3 cells were enriched in epithelial cell development and differentiation and wound healing (*Figure 2D*). In order to know whether these C3 cells represents the previously reported pro-fibrotic abnormal cells (*Wang et al., 2023*), we compared their transcriptomic profile with the basal cells isolated from the injured area of IPF lungs (*Heinzelmann et al., 2022*). The data showed that in comparison to the C3 cells isolated from healthy area, the basal cells isolated from injured area were significantly enriched with genes related to DNA damage, senescence, apoptosis, as well as genes related to the differentiation of mesenchymal cells and fibroblasts (*Figure 1—figure supplement 2C*). Therefore, the C3 cells are different from those pro-fibrotic basal cells. Altogether these data indicated that the current cell

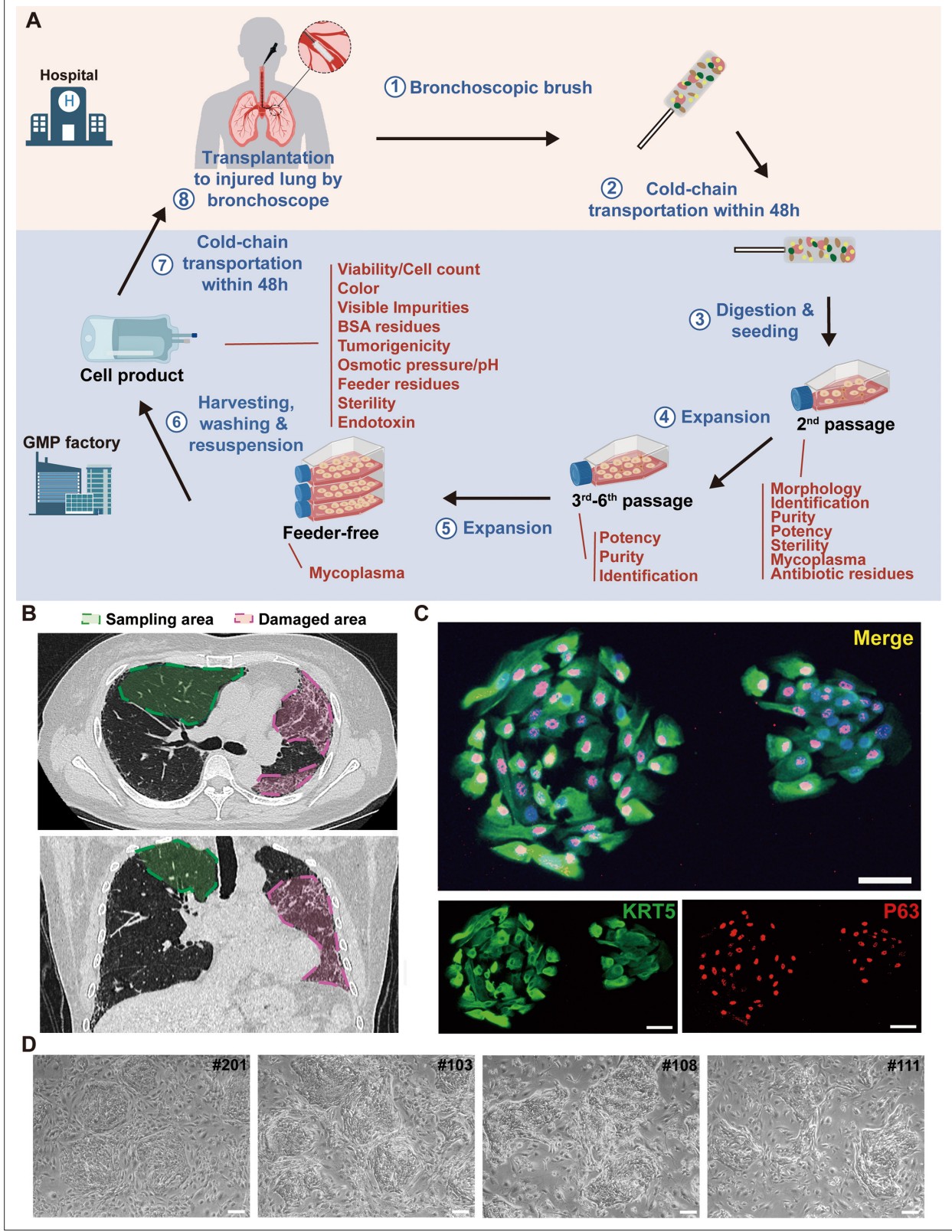

**Figure 1.** Cloning and characterization of P63+ lung progenitor cells isolated from IPF patients. (**A**) A schematic diagram illustrating the manufacturing, quality control, and clinical administration of autologous P63+ lung progenitor cell product REGEND001. (**B**) Representative lung CT images of sampling and damaged areas of the IPF patients. Green circle: sampling area; red circle: damaged area. (**C**) Immunostaining of clonogenic cells with basal cell

*Figure 1 continued on next page*

*Figure 1 continued*

markers KRT5 (green) and P63 (red). Scale bar, 30 µm. (**D**) Representative images of cell clones showing distinct morphology from different individual patients (indicated by patient numbers). Scale bar, 80 µm.

The online version of this article includes the following figure supplement(s) for figure 1:

**Figure supplement 1.** Profile of current clinical trial.

**Figure supplement 2.** Quality control of cultured lung progenitor cells.

isolation and expansion system could efficiently enrich normal P63+ lung progenitor cells from IPF patients.

At final passage, P63+ progenitor cells were cultured under feeder-free conditions until reaching 85–100% confluence (*Figure 1—figure supplement 2D*). After harvesting P63+ progenitor cells using xeno-free TrypLE, cells were suspended in 14 mL of preservation reagent. The final product was sealed in cell preservation bags and shipped fresh to three clinical research centers via cold chain transportation (2–8°C) within 48 hr. Just before transplantation, cell suspension was warmed to room temperature, further diluted in saline, and then transplanted into bronchial segments of the middle and lower lobes by bronchoscope for 3 mL per segment. After the transplant surgery, the patients were asked to assume a supine position and breathe deeply for two hours to promote the adhesion of the transplanted P63+ progenitor cells to the injured lung area.

## Safety analysis of REGEND001

Patients were followed up at baseline, 1 week, 4 weeks, 12 weeks, and 24 weeks post REGEND001 treatment for safety evaluation. Until 24 weeks after transplantation, totally 62 cases of grade I-III adverse events (AEs), but no grade IV and grade V AEs, were recorded (*Table 2*). The most common AE is COVID-19. COVID-19 were recorded in all 6 patients in 2 M and 3.3 M dose groups, which were not related to cell therapy but due to the outbreak of pandemic in China from December 2022 to March 2023. 22.6% (14/62) AEs were considered related to the cell therapy. Fever is one of the mostly common cell therapy-related AEs (21.4%, 2 grade II, 2 grade I), which all happened within 24 hr after REGEND001 treatment and recovered 4–9 hr afterwards. Bloody sputum is another common cell therapy-related AEs (21.4%, all grade I), which all happened within 24 hr after REGEND001 treatment and recovered within 1–3 days. The reason for bloody sputum was considered to be related to the bronchoscopic surgery but not the cell product. The grade III AEs include 2 severe COVID-19 and 2 acute bronchitis. The 2 acute bronchitis happened in two patients in 2 months and 4 months post cell transplantation respectively, and both recovered after standard treatment. Considering the medical history of the two patients, we think the acute bronchitis were not related to the cell treatment.

The results of blood routine and blood biochemistry in all patients were plotted in *Appendix 1—figure 1* and *Appendix 1—figure 2*. Overall, majority of these measures were maintained within the normal reference ranges. For the abnormal readings, most of them were clinically meaningless. The most common clinically meaningful abnormality is increasement of blood glucose level in five patients with diabetes. In addition, mild increase of eosinophil level was observed in 1 M group. By consecutive high-resolution CT (HRCT) scanning of the chest, we did not find any signs of malignancy or new pathologies in any patient. Consistently, no clinically meaningful change of 4 tumor markers in blood sample was observed (*Appendix 1—figure 3*). 12-lead Electrocardiogram (ECG) and urine routine results were also normal after cell treatment. Altogether these data indicated that autologous P63+ progenitor cells transplantation therapy had an acceptable safety profile among IPF patients.

## Change of lung functions after REGEND001 treatment

One key efficacy outcome of the current study is the change in gas transfer function after cell therapy. Lung gas transfer function is determined by effective alveolar capillary surface area, which could be quantitatively measured by diffusion capacity of lung for carbon monoxide (DLCO) and alveolar volume-corrected DLCO (DLCO/VA). In patients with IPF, progression of pulmonary fibrosis is the primary cause of decline in gas transfer function, which is a consistent and strong predictor of mortality in patients with various fibrotic lung diseases (*Raghu et al., 2022*). In current trial, we found that the 7 patients in the higher dose groups (1.0~3.3 M) showed statistically significant increase of DLCO/VA level trend from averagely 72.06% (baseline) to 84.10% (24 weeks) of predicted value

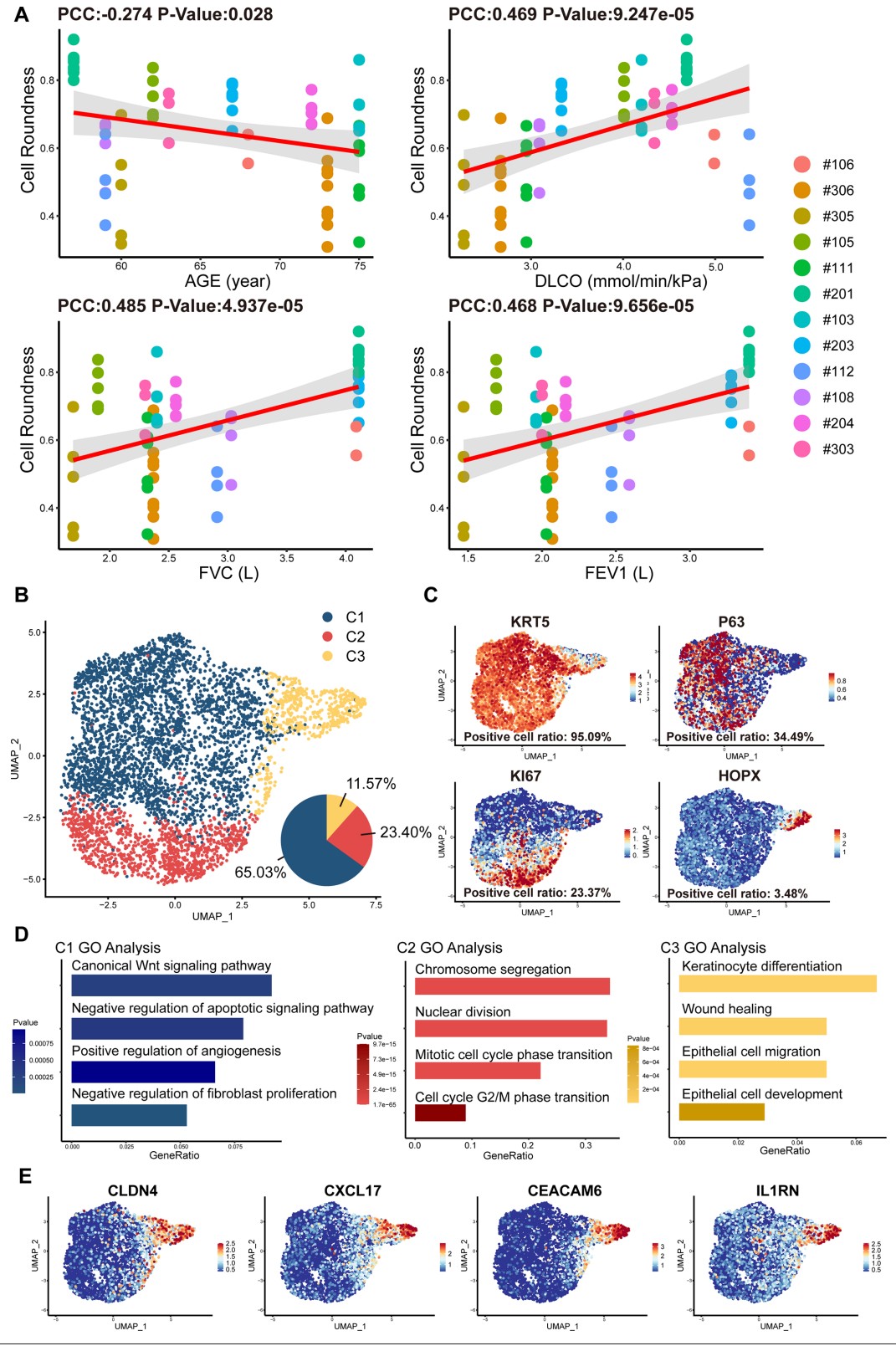

**Figure 2.** Cell morphology and single-cell RNA sequencing analysis of P63+ lung progenitor cells isolated from IPF patients. (**A**) Correlation analysis of cell colony roundness with patient age and lung functions. Each point represents one single cell colony, and each color represents an individual patient. (**B**) Uniform manifold approximation and projection (UMAP) plot showing three subpopulations of cells cloned from a patient.

*Figure 2 continued on next page*

*Figure 2 continued*

(**C**) Feature plots of representative cell markers. (**D**) Gene ontology enrichment analysis of the differentially expressed genes identified in three subpopulations. (**E**) Feature plots of representative variant progenitor cell markers.

(p-value = 0.008). Of note, the overall change of DLCO/VA in the higher dose group was statistically significant from that in the lower dose group (p-value = 0.019), indicating a dose-dependent therapeutic effect (***Figure 3A***, ***Figure 3—figure supplement 1***). The results were similar when analyzing the DLCO/VA absolute value, DLCO absolute value as well as DLCO percentage (%) to its predicted values (***Figure 3B***, ***Figure 3—figure supplement 1***). Interestingly, based on previous morphological studies, this may have suggested that variations in cell roundness were not significantly associated with changes in DLCO and DLCO/VA following treatment. Altogether, above data indicated that REGEND001 treatment had potential to improve lung gas transfer function when applied at higher dose. Moreover, we also noticed that the concomitant COVID-19 events happened in 2 M and 3.3 M group post cell treatment, which might compromise some the potential benefit from REGEND001 treatment (***Figure 3—figure supplement 1***).

We also analyzed the change of forced vital capacity (FVC) after REGEND001 treatment. FVC represents the airflow aspect of lung function, and its decline is the indicator of the progression of IPF (***Fainberg et al., 2022***). Unlike DLCO, in the higher dose groups, we only observed minimal improvement of FVC% from averagely 76.91% (baseline) to 77.01% (24 weeks). The overall change of FVC in the higher dose group was statistically significant from that in the lower dose group (p=0.028; ***Figure 3C***, ***Figure 3—figure supplement 1***). In the higher dose groups, three of seven patients demonstrated >2% positive change of FVC% after REGEND001 treatment, which reached the minimal clinically important difference (MCID) for FVC% change in IPF patients (***du Bois et al., 2011a***). Altogether, these data suggested that REGEND001 treatment might have potential to stabilize or slightly improve FVC when given at a high dose.

**Table 2.** Adverse events[*].

| | No. (%) of Patients With Adverse Events | | | |
|---|---|---|---|---|
| **Events** | **0.6 M** | **1M** | **2 M** | **3.3 M** |
| Any adverse event | 2 (16.67) | 3 (25.00) | 3 (25.00) | 3 (25.00) |
| Serious adverse events | 2 (16.67) | 0 | 1 (8.33) | 1 (8.33) |
| Fatal adverse events | 0 | 0 | 0 | 0 |
| Adverse events leading to discontinuation | 0 | 0 | 0 | 0 |
| Frequent adverse events:[†] | | | | |
| Likely related to Bronchoscopy | | | | |
| Hemoptysis | 1 (8.33) | 1 (8.33) | 1 (8.33) | 1 (8.33) |
| Fever | 0 | 1 (8.33) | 2 (16.67) | 1 (8.33) |
| White blood cell counts increased | 0 | 1 (8.33) | 0 | 1 (8.33) |
| Productive cough | 1 (8.33) | 1 (8.33) | 0 | 0 |
| Bronchitis | 1 (8.33) | 0 | 0 | 0 |
| Others | | | | |
| COVID-19 | 0 | 0 | 3 (25.00) | 3 (25.00) |
| Bronchitis | 2 (16.67) | 0 | 0 | 1 (8.33) |
| Hypokalemia | 1 (8.33) | 0 | 1 (8.33) | 0 |

[*]Shown are adverse events occurred in patients from baseline examination to the end of the study visit. Events are listed in descending order of frequency.

[†]The frequent adverse events were defined as those with an incidence ≥2 patients, which were ordered by frequency of occurrence here.

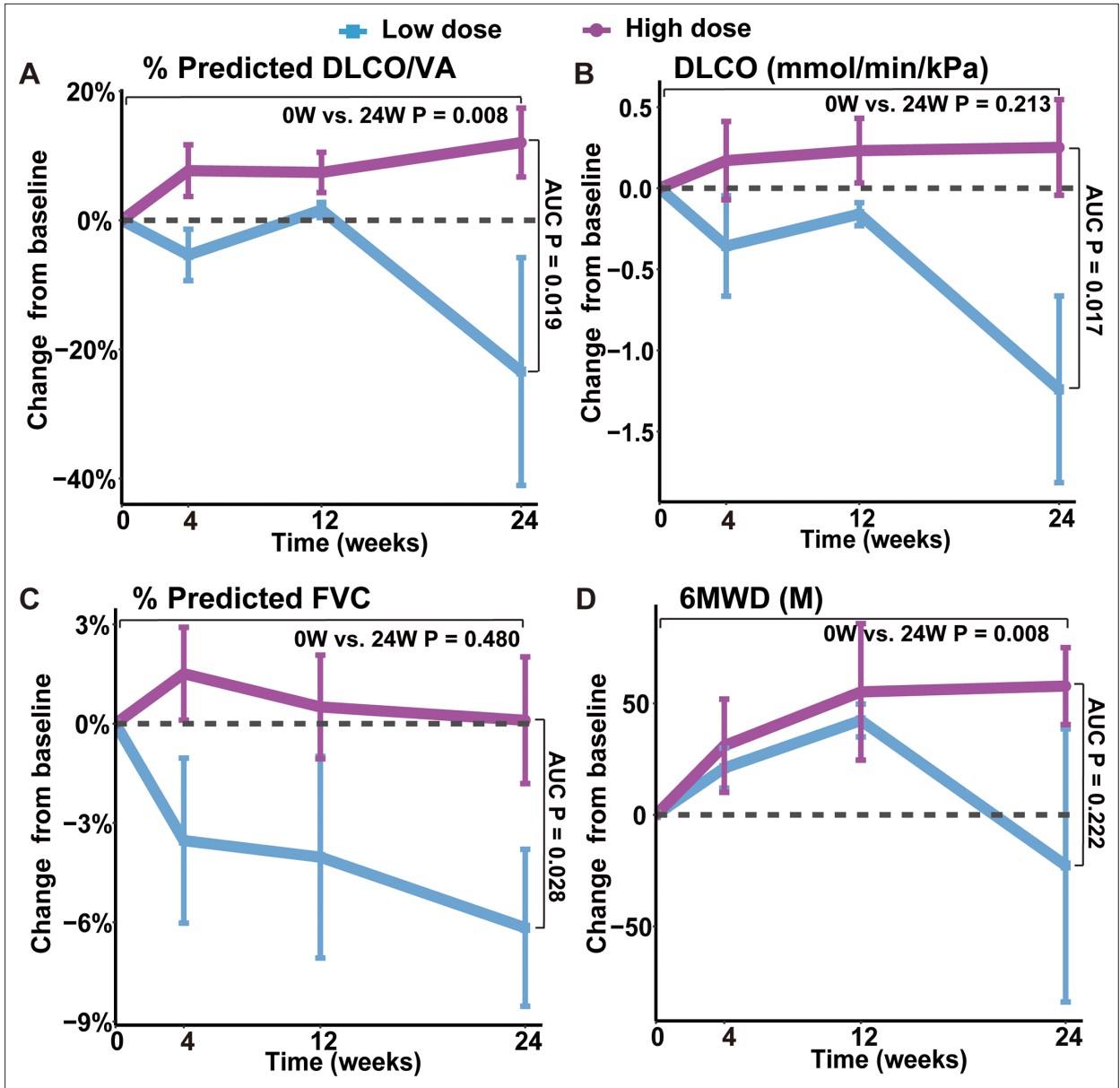

**Figure 3.** Changes of lung functions and 6MWD at different time points after REGEND001 treatment. (**A–D**) The plots indicated mean (S.E.M.) changes from baseline in absolute value of DLCO/VA percentage to predicted value, DLCO absolute value, FVC percentage to predicted value and 6MWD. The low dose group included the 3 patients in the 0.6 M group, while the high dose groups included the 7 patients in the 1 M, 2 M, and 3.3 M dose groups. p Values between two groups were calculated according to the area under curve (AUC) and are labeled as 'AUC P'. p Values for the change from baseline to 24 weeks within high dose group are labeled as '0 W vs. 24 W P'.

The online version of this article includes the following figure supplement(s) for figure 3:

**Figure supplement 1.** Changes of predicted DLCO%, DLCO, predicted DLCO/VA%, DLCO/VA, predicted FVC%, FVC, 6MWD and SGRQ score at different time points after cell therapy.

## Change of exercise ability, quality of life and lung CT image

We also analyzed the 6 min walking distance (6MWD) of patients, which is a widely used measure to assess functional exercise ability. In consistent with their improvement of lung functions, patients in higher dose groups demonstrated statistically significant improvement of 6MWD from averagely 424 m at baseline to 482 m at 24 weeks (p-value = 0.008; *Figure 3D*, *Figure 3—figure supplement 1*). The 58 m increase of 6MWD is considered clinically quite meaningful (*du Bois et al., 2011c*). We also evaluated the patients' quality of life by St. George's Respiratory Questionnaire (SGRQ), which

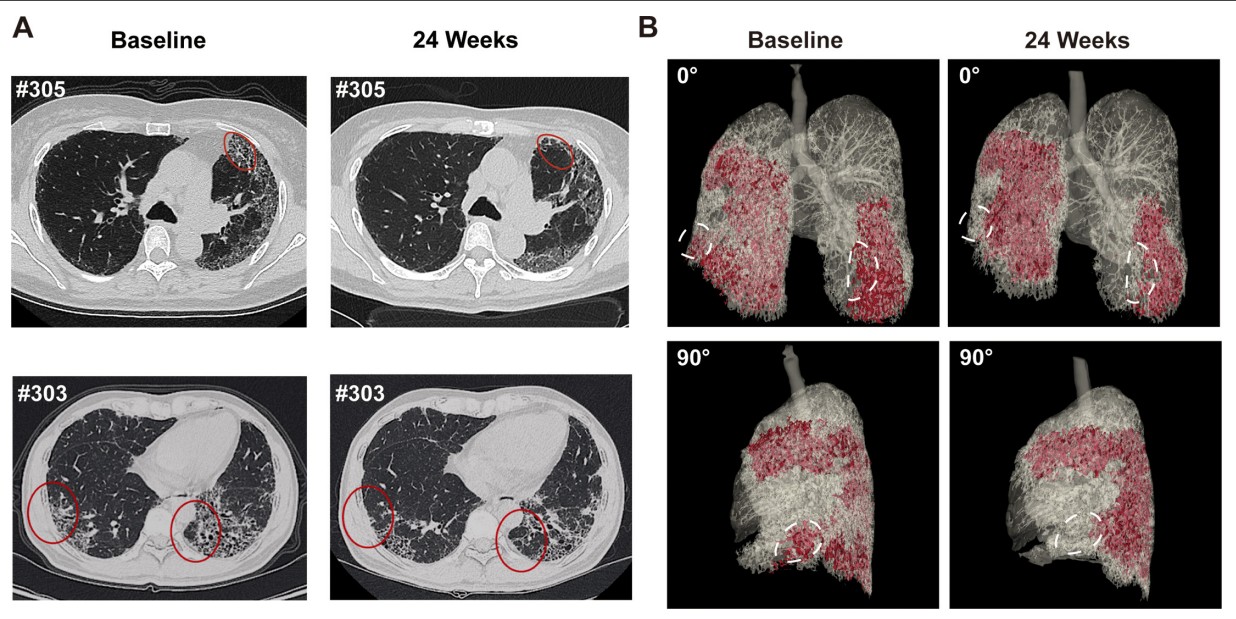

**Figure 4.** Representative lung CT image before and after REGEND001 treatment. (**A**) Representative lung CT images of the Patient #305 and #303 at baseline and 24 weeks after REGEND001 treatment. Red circle indicated resolution of honeycomb lesion. (**B**) Three-dimensional visualization of consecutive CT images of the Patient #305. The red zone indicated the lung damaged areas (reticulation and honeycomb) before and after cell therapy. The white circle indicated resolution of the lesion in lower lobes.

was originally developed and validated in obstructive airway disease and has acceptable validity and reliability in IPF patients (***Prior et al., 2019***). Its scores range from 0 to 100, with higher scores indicating more limitations in the overall health of patient, daily life, and perceived well-being. Similar to the 6MWD result, patients in higher dose groups demonstrated decrease of SGRQ numerically from mean 32.3 at baseline to 25.8 at 24 weeks (***Figure 3—figure supplement 1***). The 6 points change of SGRQ score were considered clinically quite meaningful (***Kang et al., 2021***). Altogether, these findings suggest that REGEND001 treatment, especially at higher dose groups, could help to improve the exercise ability and quality of life in IPF patients.

We also used HRCT scan to evaluate changes in the lung morphology of patients. In IPF lung, honeycombing cyst was a characteristic structure, which was developed after collapse of fibrotic alveolar wall and dilatation of terminal airways. The honeycombing pattern in CT imaging is considered permanent and independently associated with the disease prognosis (***Raghu et al., 2022***). In current study, three experts independently evaluated the lung CT images at baseline and 24 weeks after REGEND001 treatment in blinded manner. The results indicated that at baseline, all recruited IPF patients shows predominantly peripheral and lower lobe bilateral reticulation and honeycombing pattern. At 24 weeks, as most patients showed no obvious change of lung CT images post cell therapy, the Patient #303 and #305 in 1 M group demonstrated resolution of honeycombing lesion in their lower lungs (***Figure 4A***). We also performed three-dimensional (3D) computational visualization of reticulation and honeycombing in #305 CT images. Interestingly, the 3D data showed that the improvement of lung lesion was exclusively observed in the lower lobes (***Figure 4B***). One explanation could be that the cell suspension delivered through the bronchoscope tend to flow into the lower lobes due to gravity, which would result in cell concentration in lower lobes. Similar effect was also observed in the repair of mild emphysematous lesion in COPD patients in our previous study (***Wang et al., 2024***). In consistency with the improvement of CT images in #303 and #305, these two patients demonstrated improvement of DLCO, FVC, and 6MWD as well (***Figure 3—figure supplement 1***). Interestingly, based on the data in ***Figure 2***, the cell cloned from #303 and #305 had vastly diverse roundness, suggesting that high clone roundness morphology might not be required for therapeutic efficacy. Altogether the above data suggested that REGEND001 treatment have potential to repair the pulmonary fibrotic structure at least in some patients in 1 M group.

## Discussion

The destruction of alveolar tissues in the lungs of IPF patients is currently irreversible. Thus, there is an urgent need for treatment options that support lung repair and/or regeneration. Previous studies have demonstrated that human P63+ lung progenitor cells can be easily isolated, expanded, and transplanted for lung regeneration (*Ma et al., 2018*; *Wang et al., 2024*). In the current study, using single-cell RNA sequencing, we characterized the cells cloned from healthy upper lobe of IPF patient. Interestingly, apart from the KRT5+/P63+ basal progenitor cells, we also identified a subpopulation of 'variant progenitor cells' in IPF patients. Similar variant cells have been found in healthy donors in low numbers as well as in the injured lobes of IPF patients with high abundance (*Wang et al., 2023*). Interestingly, our cells cloned from healthy lobe have a low abundance of variant cells exactly like those cloned from healthy donors (*Wang et al., 2023*). Moreover, unlike the previously reported pro-fibrotic subpopulation in IPF patients, our variant progenitor cells exhibited a higher expression level of the HOPX gene, which is known to be expressed in normal type I alveolar cells, while exhibited lower expression of fibrosis-related genes (*Heinzelmann et al., 2022*). These data suggest that we can clone relatively normal, but not pro-fibrotic, progenitor cells from the healthy lobes of IPF patients, which can be used to manufacture a qualified cell therapeutic product.

Some previous studies also proposed that those abnormal, dysplastic P63+ progenitor cells located in the injured lung area of IPF patients could halt the repair process, participate in 'honeycomb' structure formation and be deleterious for the lung function (*Jaeger et al., 2022*; *Hewitt et al., 2023*). These findings raised 'red flag' for the safe transplantation of P63+ progenitor cells in IPF patients. To address the safety issue of autologous P63+ progenitor cells, we conducted the current dose-escalation phase 1 clinical trial. Our data indicated minimal safety issues, with all events related to cell administration being mild and transient. Considering our previous pre-clinical animal data altogether (*Zhou et al., 2021*), we now know that at least the P63+ progenitor cells isolated from the healthy lobes of IPF patients are safe for intrapulmonary transplantation.

Although the trial was primarily focused on safety and not designed to assess efficacy, the data obtained also suggested meaningful improvements in lung gas transfer function DLCO, which is known to be closely related to the progression of pulmonary fibrosis. DLCO is considered to be a consistent and powerful predictor of mortality in patients with various fibrotic lung diseases including IPF (*Raghu et al., 2022*). Low DLCO level are strongly associated with diminished survival rate (*Latsi et al., 2003*; *Fell et al., 2009*; *du Bois et al., 2011b*; *Ley et al., 2011*; *Song et al., 2019*). A drop of more than 15% in DLCO was associated with a threefold increased risk of death compared to a drop of less than 15% in IPF patients (*Doubková et al., 2018*; *Collard et al., 2003*). Additionally, DLCO also has a predictive effect on the exercise ability of IPF patients (*Fernández Fabrellas et al., 2018*; *Lei et al., 2022*; *Smyth et al., 2023*; *Silva et al., 2024*).

Earlier studies indicated that conventional anti-fibrotic drugs could not achieve improvements in DLCO in IPF patients. In a clinical trial of conventional IPF treatment, Durheim et al. analyzed data from 436 IPF patients after initiating antifibrotic therapy and reported an annual change rate of predicted DLCO% of –2.9% (*Durheim et al., 2021*). Similarly, when Zulkova et al. evaluated the effect of pirfenidone on lung function decline and survival based on 383 patients with IPF, they found that the predicted mean change in DLCO% at week 24 was –1.6% (*Zurkova et al., 2019*). In contrast, our study demonstrated a+12.0% increase in DLCO/VA% and a+5.4% increase in DLCO% from baseline at 24 weeks in higher dose groups. Although there were data from only 7 patients in the higher dose groups, it still implies a potential advantage over conventional therapy, showing a statistically significant difference from patients in the lower dose group.

The current study has several limitations that need to be addressed in future research. Firstly, the sample size of the study was very small. Secondly, the absence of a placebo-controlled arm makes it difficult to draw firm conclusions about the efficacy of the treatment. Thirdly, the progenitor cells cloned from individual patients had their own morphological properties as well as genetic and epigenetic background, which might lead to distinct response to therapy in different individuals. Fourthly, the data collection and analysis of the 2 M and 3.3 M patient groups were interfered by the concomitant COVID-19 events and city lockdown policy. To overcome some of these limitations, a randomized, placebo-controlled, blind, multi-center phase 2 study with a larger sample size is currently being performed in China (Clinical Trial No. NCT06081621). Hopefully, the phase 2 trial might address the limitations of the current study and provide more reliable evidence for the efficacy of the treatment.

## Acknowledgements

This study was supported by National High Level Hospital Clinical Research Funding (2022-PUMCH-B-108 to Z-JX), National Key Research and Development Plan (2024YFA1108900 to TZ and 2024YFA1108500 to WZ), Jiangsu Province Science and Technology Special Project (Key Research and Development Plan for Social Development) Funding (BE2023727 to WZ), National Biopharmaceutical Technology Research Project Funding (NCTIB2023XB01011 to WZ) and Non-profit Central Research Institute Fund of Chinese Academy of Medical Science (2020-PT320-005 to WZ). Regend Therapeutics also funded the study. The authors also thank the patients who participated in this study and their families, and the study coordinators. We thank Jie Ren for research assistance.

## Additional information

### Competing interests

Ting Zhang, Wei Zuo: Regend Therapeutics is holding the patent of human lung progenitor cell isolation and expansion technique. The other authors declare that no competing interests exist.

### Funding

| Funder | Grant reference number | Author |
|---|---|---|
| National High Level Hospital Clinical Research Funding | 2022-PUMCH-B-108 | Zuojun Xu |
| National Key Research and Development Plan | 2024YFA1108900 | Ting Zhang |
| National Key Research and Development Plan | 2024YFA1108500 | Wei Zuo |
| Jiangsu Province Science and Technology Special Project (Key Research and Development Plan for Social Development) Funding | BE2023727 | Wei Zuo |
| National Biopharmaceutical Technology Research Project Funding | NCTIB2023XB01011 | Wei Zuo |
| Non-profit Central Research institute Fund of Chinese Academy of Medical Science | 2020-PT320-005 | Wei Zuo |

The funders had no role in study design, data collection and interpretation, or the decision to submit the work for publication.

### Author contributions

Shiyu Zhang, Yu Zhao, Formal analysis, Investigation, Visualization, Writing – original draft, Writing – review and editing; Min Zhou, Chi Shao, Lei Ni, Zhiyao Bao, Qiurui Zhang, Resources, Investigation, Writing – review and editing; Mingzhe Liu, Formal analysis, Writing – review and editing; Ting Zhang, Data curation, Supervision, Funding acquisition, Writing – original draft, Project administration, Writing – review and editing; Qun Luo, Jieming Qu, Resources, Supervision, Validation, Investigation, Methodology, Project administration, Writing – review and editing; Zuojun Xu, Resources, Supervision, Funding acquisition, Validation, Investigation, Methodology, Project administration, Writing – review and editing; Wei Zuo, Conceptualization, Supervision, Funding acquisition, Writing – original draft, Project administration, Writing – review and editing

### Author ORCIDs

Wei Zuo ⓘ https://orcid.org/0000-0002-4460-0337

## Ethics

Clinical trial registration Trial Registration: Chinadrugtrials.org; Identifier: CTR20210349.

Human subjects: Both the informed consent and the ethical approval were obtained. Approving organization: The Medical Ethics Committee of the First Affiliated Hospital of Guangzhou Medical University; Ruijin Hospital Ethics Committee, Shanghai Jiao Tong University School of Medicine; Drug Clinical Trial Ethics Committee of Peking Union Medical College Hospital, Chinese Academy of Medical Sciences. Approval number: CXSL1900019. The specific Informed Consent Forms and Ethics Committee Approval content have been uploaded in the Supplementary files.

Reviewer #1 (Public review): https://doi.org/10.7554/eLife.102451.3.sa1
Author response https://doi.org/10.7554/eLife.102451.3.sa2

---

## Additional files

### Supplementary files

MDAR checklist

Supplementary file 1. CTR20210349 (English Version).

Supplementary file 2. Ethics Committee Approval (English Version).

Supplementary file 3. Informed consent forms (English Version).

Supplementary file 4. Protocol for IPF Clinical Trial.

### Data availability

Sequencing data have been deposited in GEO under accession codes GSE269794. The single-cell RNA sequencing data of distal lungs of IPF patients was obtained from GSE190889. The majority of the analysis was carried out using published and freely available software and pre-existing packages mentioned in the methods. No custom code was generated.

The following dataset was generated:

| Author(s) | Year | Dataset title | Dataset URL | Database and Identifier |
|---|---|---|---|---|
| Zhao Y | 2024 | Single cell RNA sequencing analysis of P63+ lung progenitor cells isolated from IPF patients | https://www.ncbi.nlm.nih.gov/geo/query/acc.cgi?acc=GSE269794 | NCBI Gene Expression Omnibus, GSE269794 |

The following previously published dataset was used:

| Author(s) | Year | Dataset title | Dataset URL | Database and Identifier |
|---|---|---|---|---|
| Heinzelmann K, Hu Q, Hu Y, Dobrinskikh E, Ansari M, Melo-Narváez M, Ulke HM, Leavitt C, Mirita C, Trudeau T, Saal ML, Rice P, Gao B, Janssen WJ, Yang IV, Schiller HB, Vladar EK, Lehmann M, Königshoff M | 2022 | Single cell RNA Sequencing Identifies G-protein Coupled Receptor 87 as a Basal Cell Marker Expressed in Distal Honeycomb Cysts in Idiopathic Pulmonary Fibrosis | https://www.ncbi.nlm.nih.gov/geo/query/acc.cgi?acc=GSE190889 | NCBI Gene Expression Omnibus, GSE190889 |

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

# Appendix 1

## List of inclusion and exclusion criteria
### Inclusion Criteria:

1. Male or female, aged between 50–75;
2. Subjects diagnosed with IPF according to guidelines for the diagnosis of idiopathic pulmonary fibrosis 2018 edition;
3. Subjects with 30%~79% of the predicted value in diffusing capacity for carbon monoxide (DLCO) and more than 50% of the predicted value in forced vital capacity (FVC) in pulmonary function tests 3 months before screening;
4. Subjects with typical high-resolution computed tomography (HR-CT) imaging findings of idiopathic pulmonary fibrosis in the past 12 months;
5. Subjects tolerant to bronchofiberscope;
6. Subjects fully informed of the purpose, method and possible discomfort of the trial, agreeing to participate in the test, and voluntarily signing the informed consent;
7. Subjects with good adherence, willingness to take medication and regular follow-up examinations as required by the protocol;
8. Subjects able to understand and cooperate with the completion of pulmonary function tests.

### Exclusion Criteria:

1. Subjects who cannot tolerate cell therapy;
2. Pregnant or lactating women;
3. Subjects with syphilis or any of human immunodeficiency virus (HIV), hepatitis B virus (HBV), hepatitis C virus (HCV) positive antibody; Of which stable HBV carriers after drug treatment (DNA titer ≤500 IU/mL or copy number <1,000 copies/mL) and cured hepatitis C patients (HCV RNA is negative) can be enrolled;
4. Subjects with malignant tumors or a history of malignant tumors;
5. Subjects with taking drugs which caused lung fibroblast such as amiodarone in a long term before screening;
6. Subjects with infections in lung or other site, including bacterial and viral infections, requiring intravenous treatment before cell transplantation;
7. Subjects with a history of invasive or noninvasive mechanical ventilation within 4 weeks;
8. Subjects with any of the following lung diseases: asthma, active tuberculosis, pulmonary embolism, pneumothorax, pulmonary hypertension, pneumoconiosis, etc.; lung cancer, bronchiolitis obliterans or other active lung disease; Pneumonia currently or within the last 4 weeks; Pneumonectomy Previously;
9. Subjects needing oxygen therapy currently (oxygen therapy time >15 h/d);
10. Subjects suffering from other serious systemic diseases, such as myocardial infarction, unstable angina, liver cirrhosis, acute glomerulonephritis, connective tissue disease, etc.;
11. Subjects with following results: leukopenia (leukopenia <4 × 10^9 /L) or agranulocytosis (leukocyte <1.5 × 10^9 /L or neutrophils <0.5 × 10^9 /L) of any cause; Blood creatinine >2.5 times the upper limit of normal; Alanine transaminase (ALT) and Aspartate transaminase (AST)>2.5 times the upper limit of normal values in the laboratory tests;
12. Subjects with a history of mental illness or suicide risk, epilepsy or other central nervous system disorders;
13. Subjects with severe arrhythmias (such as ventricular tachycardia, frequent supraventricular tachycardia, atrial fibrillation, atrial flutter, etc.) or atrioventricular block of degree II or above, shown by 12-lead Electrocardiogram (ECG);
14. Subjects with a history of abusing alcohol and illicit drug;
15. Subjects who are allergic to cattle products;
16. Subjects who participated in other clinical trials in the past 3 months;
17. Subjects with poor compliance and difficult to complete the investigation;
18. Investigators, employees of research centers or family members of them (none of whom are suitable to participate in the trial to ensure the objectivity of the research);

19. Subjects who had an acute exacerbation of IPF or hospitalized for other respiratory diseases 3 or more times in the past 1 year;
20. Subjects who take nintedanib for medication within 1 month, or plan to continue taking nintedanib for medication;
21. Subjects with other acquired or congenital immunodeficiency disorders, or with a history of organ transplantation or cell transplant therapy;
22. Subjects whose expected survival may be less than one year judged by the investigator;
23. Male participants of childbearing potential and female participants within childbearing age were reluctant to use effective contraception from the time of signing the informed consent to 6 months after cell therapy;
24. Subjects assessed as inappropriate to participate in this clinical trial by investigator.

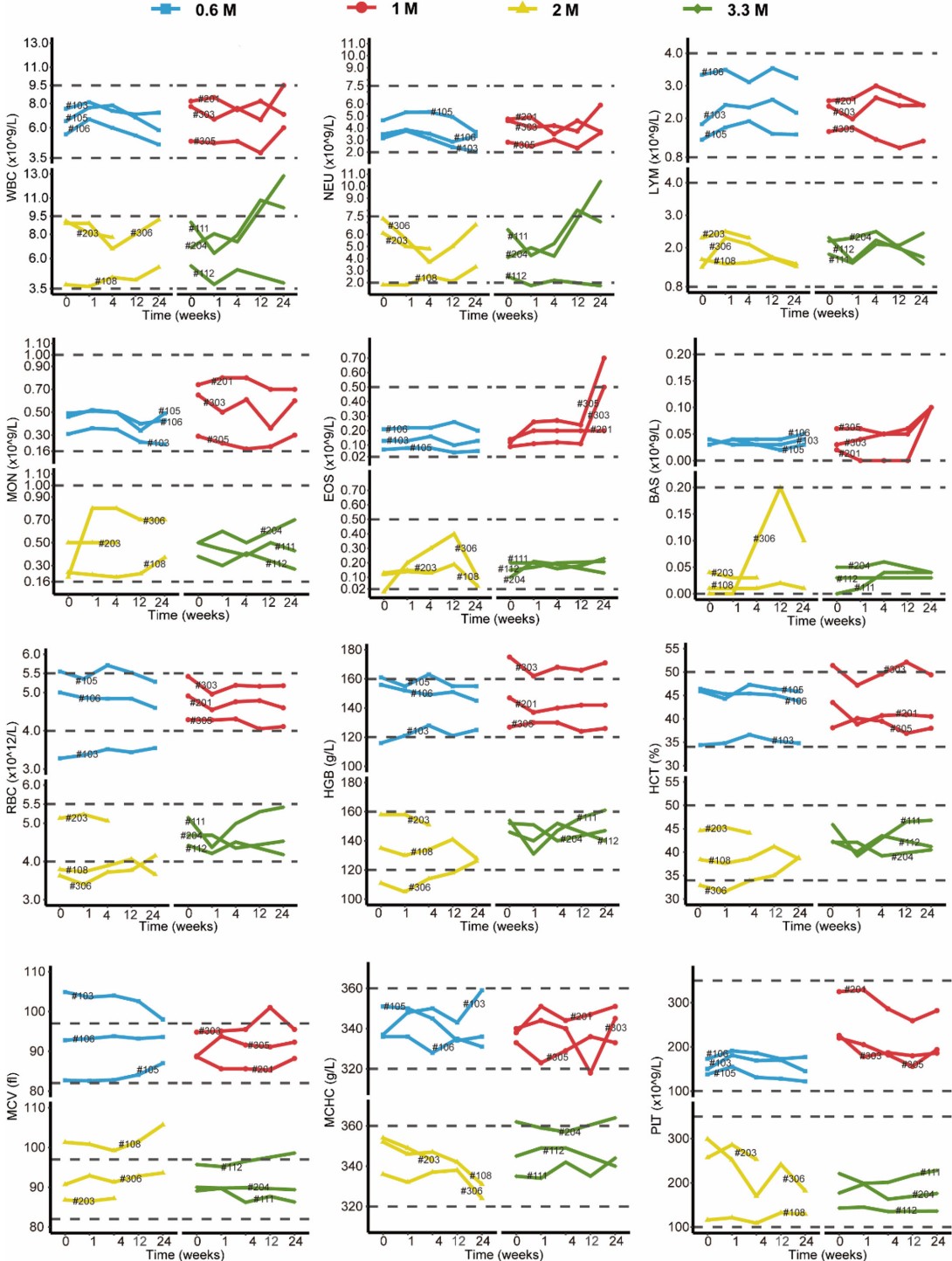

**Appendix 1—figure 1.** Changes in clinical laboratory evaluations of blood routine following cell therapy. The line plot displays the changes of various indicators of blood routine for patients over time, with each line representing one patient, and the patient number is marked alongside the line. The horizontal dashed line indicates the normal reference range.

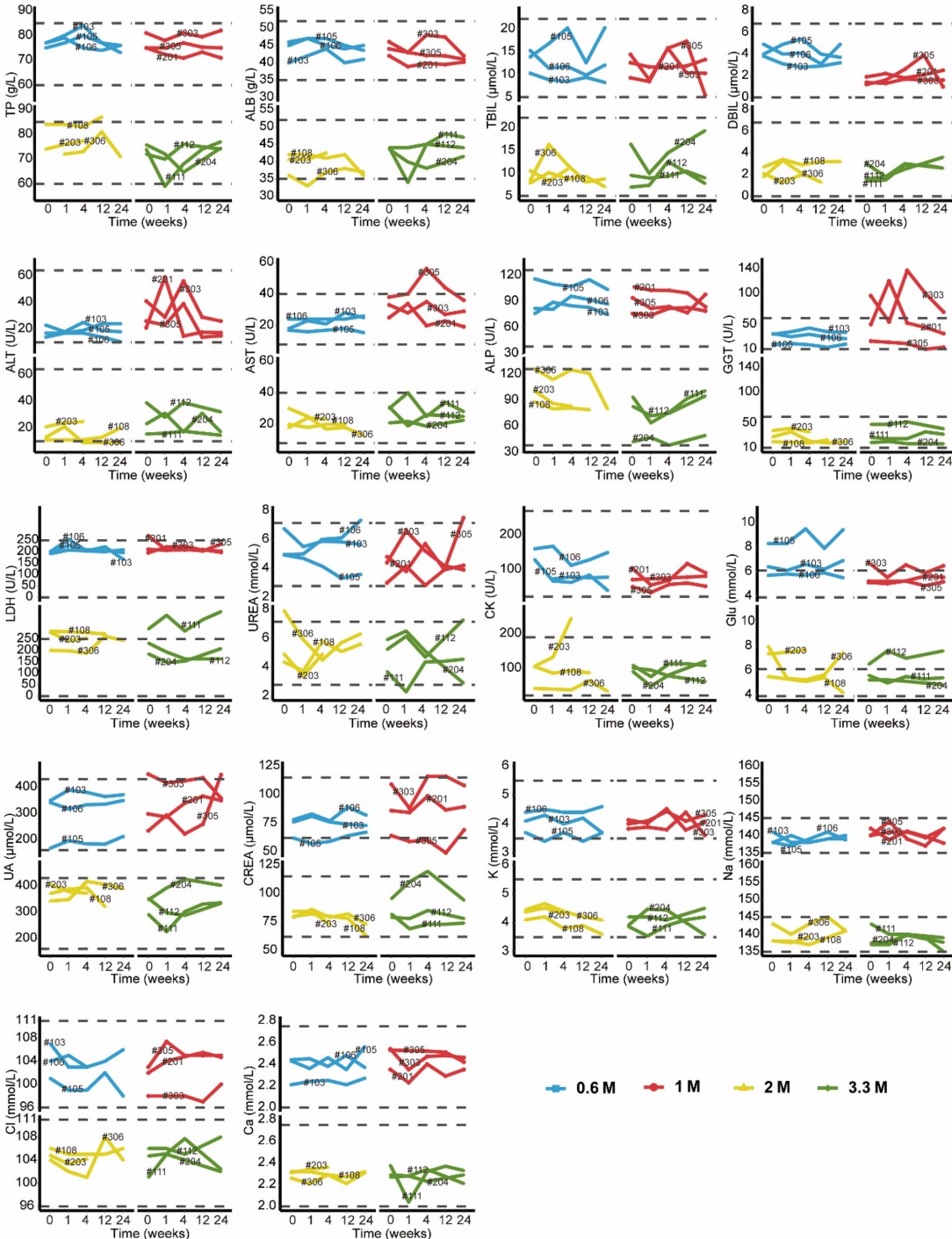

**Appendix 1—figure 2.** Changes in clinical laboratory evaluations of blood biochemistry following cell therapy. The line plot displays the changes of various blood biochemistry indicators over time, with each line representing a patient, and the patient number is labeled alongside the line. The horizontal dashed line indicates the normal reference range.

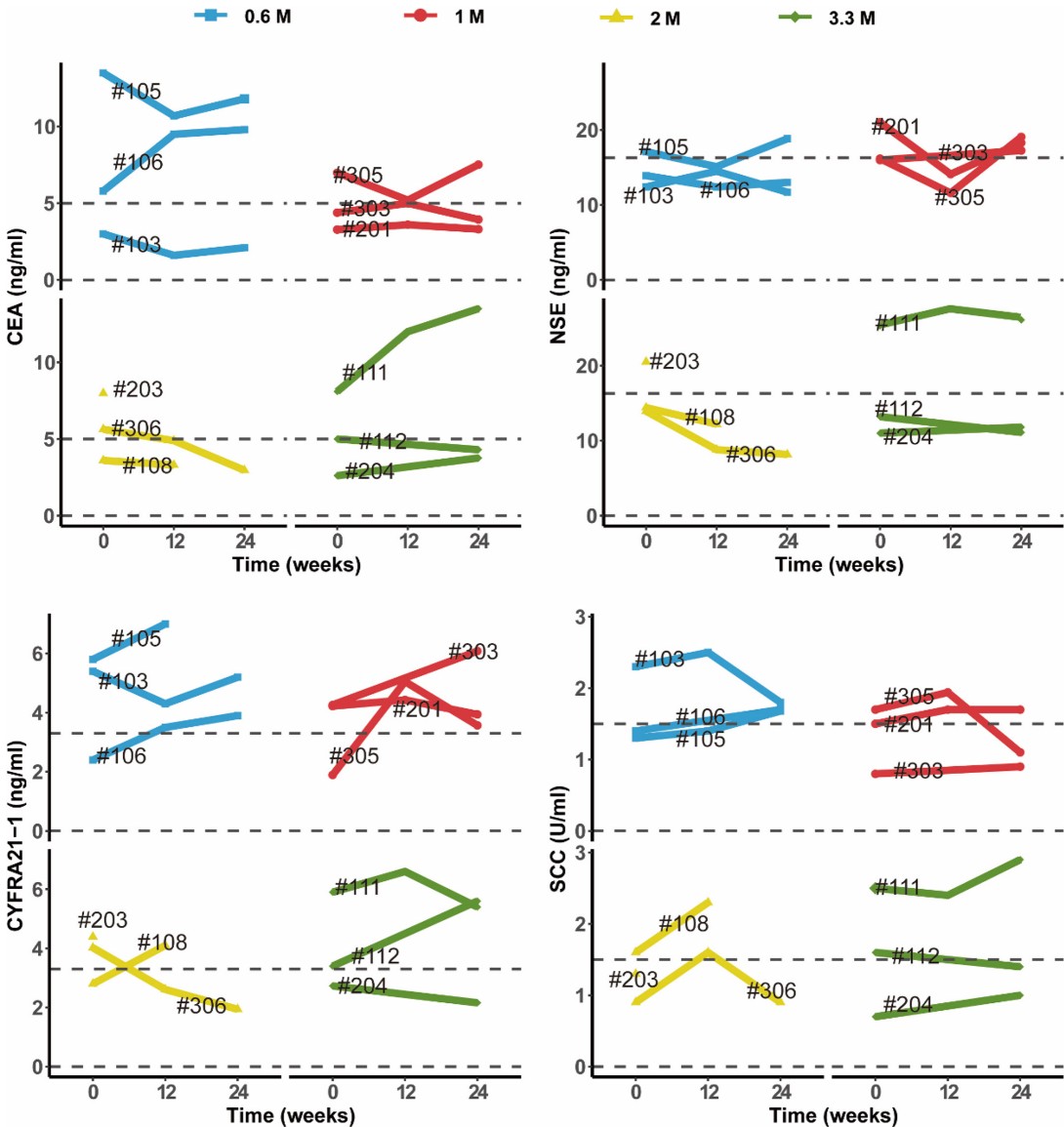

**Appendix 1—figure 3.** Changes in clinical laboratory evaluations of serum tumor markers following cell therapy. The line plot displays the changes of various tumor markers in blood sample of patients over time, with each line representing one patient, and the patient number is labeled alongside the line. The horizontal dashed line indicates the normal reference range for that indicator.

