## [Editor Report · eLife Assessment]

This **important** study describes a first-in-human trial of autologous p63+ stem cells in patients with idiopathic pulmonary fibrosis, a lethal condition for which effective treatments are lacking. The authors provide **convincing** evidence that P63+ progenitor cell therapy can be safely delivered in patients with ILD, warranting movement to a Phase 2. However, given that this is a Phase 1 study with a small sample size, conclusions regarding efficacy should not yet be made.

---

## [Referee Report · Reviewer #1 (Public review)]

Summary:

IPF is a disease lacking regressive therapies which has a poor prognosis, and so new therapies are needed. This ambitious phase 1 study builds on the authors 2024 experience in Sci Tran Med with positive results with autologous transplantation of P63 progenitor cells in patients with COPD. The current study suggests P63+ progenitor cell therapy is safe in patients with ILD. The authors attribute this to acquisition of cells from a healthy upper lobe site, removed from the lung fibrosis. There are currently no cell based therapies for ILD and in this regard the study is novel with important potential for clinical impact if validated in Phase 2 and 3 clinical trials.

Strengths:

This study addresses the need for an effective therapy for interstitial lung disease. It offers good evidence the cell used for therapy are safe. In so doing it addresses a concern that some P63+ progenitor cells may be proinflammatory and harmful, as has been raised in the literature (articles which suggested some P63+ cells can promote honeycombing fibrosis; ref 26 & 35). The authors attribute the safety they observed (without proof) to the high HOPX expression of administered cells (a marker found in normal Type 1 AECs). The totality of the RNASeq suggests the cloned cells are not fibrogenic. They also offer exploratory data suggesting a relationship between clone roundness and PFT parameters (and a negative association of patient age and clone roundness).

Weaknesses:

The authors can conclude they can isolate, clone, expand and administer P63+ progenitor cells safely; but with the small sample size and lack of placebo group no efficacy should be implied.

Comments on revisions:

The paper is meritorious as I noted initially

However, the authors did not directly address several of my concerns-i.e. their responses to the initial review were polite but did not translate into much change in the manuscript.

(1) Do these progenitor cells exert their beneficial effects by a paracrine mechanism vs transforming into lung AECs? Based on work in the field of bronchopulmonary dysplasia I suspect the benefits are mediated by a paracrine mecahnism and arguably media from these cells should be tested as an alternative to administering the cells themselves. In any case, for the revision a Discussion of the possibility of differentiation vs paracrine mechanisms, citing relevant literature, would be expected. I suggest that you add such a paragraph to a limitation section.

(2) Please address that potential implications of the fact that 5 patients had essentially normal DLCO/VA values. Saying that the "criterion for entry was DLCO" does not take away from the fact that DLCO/VA is a valid measure of lung diffusion capacity. In the absence of placebo an enrollment of mildly diseased patients would favor positive results (including stability in study endpoint parameters even without treatment). Thus, I suggest again that the limitations section should be more forthright in this regard.

(3) The authors acknowledge the lack of a placebo group but in a study of mild IPF, I worry that without a placebo group the only robust findings are those related to technique of transplantation and the safety of cell therapy. The paper still reads as if there is a clinical benefit...I would advise you further soften this (while understanding the desire to emphasize a hopeful observation). The price for not having a placebo group must be avoidance of claims of efficacy. The improvements in DLCO and CT in several cases speaks for the need for the planned phase 2 trial, which if positive will be the time to claim efficacy signals.

---

## [Author Response]

The following is the authors’ response to the original reviews.

**Reviewer #1 (Public review):**
Summary:IPF is a disease lacking regressive therapies which has a poor prognosis, and so new therapies are needed. This ambitious phase 1 study builds on the authors' 2024 experience in Sci Tran Med with positive results with autologous transplantation of P63 progenitor cells in patients with COPD. The current study suggests that P63+ progenitor cell therapy is safe in patients with ILD. The authors attribute this to the acquisition of cells from a healthy upper lobe site, removed from the lung fibrosis. There are currently no cell-based therapies for ILD and in this regard the study is novel with important potential for clinical impact if validated in Phase 2 and 3 clinical trials.Strengths:This study addresses the need for an effective therapy for interstitial lung disease. It offers good evidence that the cells used for therapy are safe. In so doing it addresses a concern that some P63+ progenitor cells may be proinflammatory and harmful, as has been raised in the literature (articles which suggested some P63+ cells can promote honeycombing fibrosis; references 26 &35). The authors attribute the safety they observed (without proof) to the high HOPX expression of administered cells (a marker found in normal Type 1 AECs). The totality of the RNASeq suggests the cloned cells are not fibrogenic. They also offer exploratory data suggesting a relationship between clone roundness and PFT parameters (and a negative association between patient age and clone roundness).

We thank the reviewer for the important comments.

Weaknesses:The authors can conclude they can isolate, clone, expand, and administer P63+ progenitor cells safely; but with the small sample size and lack of a placebo group, no efficacy should be implied.

We thank the reviewer for the suggestion and agree that we should be more cautious to discuss the efficacy of current study.

Specific points:(1) The authors acknowledge most study weaknesses including the lack of a placebo group and the concurrent COVID-19 in half the subjects (the high-dose subjects). They indicate a phase 2 trial is underway to address these issues.

N/A

(2) The authors suggest an efficacy signal on pages 18 (improvement in 2 subjects' CT scans) and 21 (improvement in DLCO) but with such a small phase 1 study and such small increases in DLCO (+5.4%) the authors should refrain from this temptation (understandable as it is).

We believe that exploring potential efficacy signal is also one aim of this study. All these efficacy endpoint analyses had been planned in prior to the start of clinical trials (as registered in ClinicalTrial.gov) and the data need be analyzed anyhow.

(3) Likewise most CT scans were unchanged and those that improved were in the mid-dose group (albeit DLCO improved in the 2 patients whose CT scans improved).

Yes, it is.

(4) The authors note an impressive 58m increase in 6MWTD in the high-dose group but again there is no placebo group, and the low-dose group has no net change in 6MWTD at 24 weeks.

Yes.

(5) I also raise the question of the enrollment criteria in which 5 patients had essentially normal DLCO/VA values. In addition there is no discussion as to whether the transplanted stem cells are retained or exert benefit by a paracrine mechanism (which is the norm for cell-based therapies).

Thank you for your detailed feedback. The enrollment criteria are based on DLCO instead of DLCO/VA. And we would like to further discuss the possible benefit by paracrine mechanism in the revised manuscript.

**Recommendations for the authors:**
(1) Four of the enrolled subjects had normal DLCO/VA (% of predicted) (>90% of predicted). This raises questions about the severity of their illness see: Table 1: Subjects 103, 105, 112, and 204 have DLCO/VA % predicted >90% of predicted and would appear not to qualify for the study. While technically enrollment criteria for DLCO are satisfied, DLCO/VA is an equally valid measure of ILD severity, and these 4 cases seem very mild.

Thank you for your detailed feedback. Yes, the current inclusion criteria is based on DLCO but not DLCO/VA. And we believe improvement of DLCO and DLCO/VA is both meaningful. In future trial, we will consider DLCO/VA as inclusion criteria as well.

(2) The authors state "Resolution of honeycomb lesion was also observed in patients of higher dose groups". This appears inaccurate as only 2 subjects in the study showed CT improvement and they were not in the highest dose group. This statement is an overreach for a Phase 1 study and should be removed from the abstract and more balance inserted in the text. The phase 2 study they are doing will answer these questions.

Thank you. We changed our statement about efficacy in the abstract part.

a) Under exclusion criteria: More detail is required as to what defines "subjects who cannot tolerate cell therapy".

Those patients cannot tolerate previous cell therapy, for example mesenchymal stem cell transplantation, would not be included in the current trial.

b) Figure S6 is important and should be in the main manuscript. This Figure shows that many (6) subjects had COVID at some trial measurement time points. This is an unfortunate confounder for efficacy signals (but efficacy is not the point of this study). Second, Figure 6 (in my view) shows little efficacy signal, which is a reminder to the authors that efficacy should not be implied in a study that was not powered to detect efficacy.

We agreed that the efficacy should be discussed very carefully.

(3) Figure S3: It appears at some does there is a significant rise in monocytes (1M cells) and neutrophils (3 M cells).

Thank you for your reasonable concerns regarding the safety of the treatment. The monocyte counts in the S3 patients, even after an increase, remains within the reference range, and therefore we consider this elevation to be clinically meaningless. One patient exhibited a significant increase in neutrophils at 24 weeks, which was attributed to a grade II adverse event, acute bronchitis, which was unrelated to cell therapy. The symptoms resolved within three days following treatment with appropriate medication.

(4) Figure 3: I wonder about the statistical significance of the 6MWD. Was this done by repeat measure ANOVA? The analysis suggests a p=0.08 but all error bars between low and high dose overlap and the biggest difference is at 24 weeks, and that appears to be labelled as not significant.

Thank you for your kind reminding. The 6MWD result with a p-value of 0.008 was derived to compare the improvement in 6MWD at the 24-week time point versus baseline within the higher group. Therefore, a paired t-test was used for this analysis. In the revised version, we label them more clearly.

**Reviewer #2 (Public review):**
Summary:This manuscript describes a first-in-human clinical trial of autologous stem cells to address IPF. The significance of this study is underscored by the limited efficacy of standard-of-care anti-fibrotic therapies and increasing knowledge of the role p63+ stem cells in lung regeneration in ARDS. While models of acute lung injury and p63+ stem cells have benefited from widespread and dynamic DAD and immune cell remodeling of damaged tissue, a key question in chronic lung disease is whether such cells could contribute to the remodeling of lung tissue that may be devoid of acute and dynamic injury. A second question is whether normal regions of the lung in an otherwise diseased organ can be identified as a source of "normal" p63+ stem cells, and how to assess these stem cells given recently identified p63+ stem cell variants emerging in chronic lung diseases including IPF. Lastly, questions of feasibility, safety, and efficacy need to be explored to set the foundation for autologous transplants to meet the huge need in chronic lung disease. The authors have addressed each of these questions to different extents in this initial study, which has yielded important if incomplete information for many of them.Strengths:As with a previous study from this group regarding autologous stem cell transplants for COPD (Ref. 24), they have shown that the stem cells they propagate do not form colonies in soft agar or cancers in these patients. While a full assessment of adverse events was confounded by a wave of Covid19 infections in the study participants, aside from brief fevers it appears these transplants are tolerated by these patients.

We thank the reviewer for the important comments.

Weaknesses:The source of stem cells for these autologous transplants is generally bronchoscopic biopsies/brushings from 5th-generation bronchi. Although stem cells have been cloned and characterized from nasal, tracheal, and distal airway biopsies, the systematic cloning and analysis of p63+ stem cells across the bronchial generations is less clear. For instance, p63+ stem cells from the nasal and tracheal mucosa appear committed to upper airway epithelia marked by 90% ciliated cells and 10% goblet cells (Kumar et al., 2011. Ref. 14). In contrast, p63+ stem cells from distal lung differentiate to epithelia replete with Club, AT2, and AT1 markers. The spectrum of p63+ stem cells in the normal bronchi of any generation is less studied. In the present study, cells are obtained by bronchoscopy from 3-5 generation bronchi and expanded by in vitro propagation. Single-cell RNA-seq identifies three clusters they refer to as C1, C2, and C3, with the major C1 cluster said to have characteristics of airway basal cells and C2 possibly the same cells in states of proliferation. Perhaps the most immediate question raised by these data is the nature of the C1/C2 cells. Whereas they are clearly p63/Krt5+ cells as are other stem cells of the airways, do they display differentiation character of "upper airway" marked by ciliated/goblet cell differentiation or those of the lung marked by AT2 and AT1 fates? This could be readily determined by 3-D differentiation in so-called airliquid interface cultures pioneered by cystic fibrosis investigators and should be done as it would directly address the validity of the sourcing protocol for autologous cells for these transplants. This would more clearly link the present study with a previous study from the same investigators (Shi et al., 2019, Ref. 9) whereby distal airway stem cells mitigated fibrosis in the murine bleomycin model. The authors should also provide methods by which the autologous cells are propagated in vitro as these could impact the quality and fate of the progenitor cells prior to transplantation.

We totally agree that the sub-population of the progenitor cells should be further analyzed. We would try this in the revised manuscript. And the methods to expand P63+ lung progenitor cells have been described in full details by Frank McKeon/Wa Xian group (Rao, et.al., STAR Protocols, 2020), which is adapted to pharmaceutical-grade technology patented by Regend Therapeutics, Ltd.

The authors should also make a more concerted effort to compare Clusters 1, 2, and 3 with the variant stem cell identified in IPF (Wang et al., 2023, Ref. 27). While some of the markers are consistent with this variant stem cell population, others are not. A more detailed informatics analysis of normal stem cells of the airways and any variants reported could clarify whether the bronchial source of autologous stem cells is the best route to these transplants.

We thank for reviewer for the good suggestion and would like to make more detailed comparison in the revised manuscript.

Other than these issues the authors should be commended for these firstin-human trials for this important condition.

Thank you so much for the kind compliment.

**Recommendations for the authors:**
Described in the review text but the authors need to be clear about how they propagated autologous stem cells in vitro.(1) Perhaps the most immediate question raised by these data is the nature of the C1/C2 cells. Whereas they are clearly p63/Krt5+ cells as are other stem cells of the airways, do they display differentiation character of "upper airway" marked by ciliated/goblet cell differentiation or those of the lung marked by AT2 and AT1 fates?

The differentiation potential of the P63+/KRT5+ basal progenitor cells have been analyzed in multiple previous literatures, which are mentioned in the revised introduction part. Basically, the human P63+ progenitor cells can differentiate into airway epithelial cells in the airway area, while give rise to immature, but functional AT1 cells in alveolar area.

(2) The authors should also provide methods by which the autologous cells are propagated in vitro as these could impact the quality and fate of the progenitor cells prior to transplantation.

The methods to expand P63+ lung progenitor cells have been described in full details by Frank McKeon/Wa Xian group (Rao, et.al., STAR Protocols, 2020), which is adapted to pharmaceutical-grade technology patented by Regend Therapeutics, Ltd.

(3) A more detailed informatics analysis of normal stem cells of the airways and any variants reported could clarify whether the bronchial source of autologous stem cells is the best route to these transplants.

We thank the reviewer for the kind suggestion and have included the comparative analysis in revised Figure S2.